# Effect of Docosahexaenoic Acid (DHA) Supplementation of Preterm Infants on Growth, Body Composition, and Blood Pressure at 7-Years Corrected Age: Follow-Up of a Randomized Controlled Trial

**DOI:** 10.3390/nu15020335

**Published:** 2023-01-10

**Authors:** Karen P. Best, Thomas R. Sullivan, Anoja W. Gunaratne, Jacqueline F. Gould, Robert A. Gibson, Carmel T. Collins, Maria Makrides, Tim J. Green

**Affiliations:** 1SAHMRI Women and Kids Theme, South Australian Health and Medical Research Institute, Adelaide, SA 5000, Australia; 2Discipline of Paediatrics, Adelaide Medical School, Faculty of Health and Medical Sciences, The University of Adelaide, Adelaide, SA 5000, Australia; 3Discipline of Public Health, Adelaide Medical School, Faculty of Health and Medical Sciences, The University of Adelaide, Adelaide, SA 5000, Australia; 4School of Agriculture Food and Wine, The University of Adelaide, Adelaide, SA 5000, Australia

**Keywords:** docosahexaenoic acid, preterm infant, growth, body composition, blood pressure, ω-3 long-chain polyunsaturated fatty acids

## Abstract

*Aim:* To determine if supplementation of infants born <33 weeks’ gestation with higher dose docosahexaenoic acid (DHA) affects growth, body composition, and blood pressure at 7 y corrected age (CA) and if treatment effects differed by infant sex at birth and birth weight strata (<1250 and ≥1250 g). *Methods:* Seven-year follow-up of an Australian multicenter randomized controlled trial in which 657 infants were fed high-DHA (≈1% total fatty acids) enteral feeds or standard-DHA (≈0.3% total fatty acids) from age 2–4 d until term CA. Seven-year CA outcomes were growth (weight, height), body composition (lean body mass, fat mass, waist, and hip circumference), and blood pressure. *Results:* There was no effect of high-DHA enteral feeds compared with standard-DHA on growth, body composition, and blood pressure at 7-year CA either overall or in subgroup analysis by sex. There was a significant interaction between high-DHA and birthweight strata on height at 7-y CA (*p* = 0.03). However, the post-hoc analyses by birthweight strata did not reach significance (*p* > 0.1). High-DHA group infants were more likely to be classified as obese (relative risk 1.6 (95% CI 1.0, 2.6); *p* = 0.05). *Conclusions:* DHA supplementation of premature infants did not affect growth, body composition, or blood pressure at 7-year CA overall by sex and birthweight strata. The finding of a higher risk of obesity in children who receive high-DHA needs to be interpreted with caution due to the small number of children classified as obese.

## 1. Introduction

The omega-3 (*n*-3) long-chain polyunsaturated fatty acid docosahexaenoic acid (DHA; 22:6 *n*-3) is the major fatty acid in the brain and is critical for neurogenesis and neurite growth [1,2]. DHA is transferred from the mother to the fetus during pregnancy, primarily during the last trimester [3,4]. Infants born prematurely are deprived of this in utero accretion of DHA, contributing to their higher risk of developmental delay compared to infants born at term [3,5]. Clinical trials assessing higher doses of DHA in preterm infants have been designed to improve neurodevelopmental outcomes [6,7,8,9,10,11]. However, DHA can be a potent anti-inflammatory agent and may have other actions that affect other organ systems. For this reason, follow-up of infants enrolled in these intervention trials is vital to assess effects on a broader range of outcomes and ensure long-term safety.

Growth failure is common in infants born prematurely. An early safety concern with supplementing preterm formula with *n*-3 fatty acids DHA and eicosapentaenoic acid (EPA) was compromised weight and length gains in early life [12,13,14], but only two studies have examined the effect of DHA supplementation on child growth and body composition beyond 24 months of age [11,15,16]. In a Scottish study, preterm infants (n = 238; born <35 weeks with birth weight < 2000 g) were randomized to receive formula supplemented with DHA (0.5%) or formula without DHA, to 9 months corrected for prematurity (corrected age [CA]) [15]. At 10 years of age, the effects on growth and blood pressure were assessed. Although there was no significant sex by treatment interaction on growth, girls who received DHA-supplemented formula were heavier and had greater skin fold thicknesses than girls who received unsupplemented formula [16]. In the largest trial of its kind, we recently reported that DHA supplementation of infants (n = 480) born before 29 weeks had no effect on weight and height z-scores at 5 years CA [11].

Preterm infants may also be at higher risk of cardiometabolic disease in later life [17]. Studies of term infants have shown that formula supplemented with *n*-3 fatty acids is associated with lower blood pressure. The only study in preterm infants showed that DHA supplementation in infancy was associated with higher systolic blood pressure at 10 years of age, but in girls only [16]

The DHA for the Improvement of Neurodevelopmental Outcome in preterm infants—The DINO Trial, is a large study to examine the role of DHA supplementation in premature infants <33 weeks of gestation (n = 657), and the children have been followed for up to 7 years. We previously assessed growth in these infants at multiple timepoints from birth to 18 months CA [14]. Infants fed higher DHA were 0·7 cm longer than infants fed standard DHA at 18 months CA. Higher DHA also resulted in increased length in infants born weighing ≥ 1250 g at 4 months CA and in both weight and length at 12 and 18 months CA. At 7-year CA there was no effect of higher DHA on average growth and body composition measures [18]. However, we have not reported whether sex or birthweight (<1250 and ≥1250 g) modify the effect of DHA on growth and body composition at 7-years CA. Moreover, we have not examined the effect of DHA on blood pressure in this group.

## 2. Materials and Methods

### 2.1. The DINO Trial

Full details of the DINO trial have been published elsewhere [6,14]. In summary, between 2001 and 2005, preterm infants born before 33 weeks gestation and their breastfeeding mother were enrolled from the neonatal units of five Australian hospitals within 5 days of commencing enteral feeds. Mother-infant pairs (n = 657) were randomly assigned to receive either a high-DHA diet (intervention) or standard-DHA diet (control) until 40 weeks postmenstrual age, using a computerized telephone randomization service. Randomization was stratified by hospital (center), birthweight (<1250 and ≥1250 g), and infant sex. Multiple births were considered a single randomization unit, and the randomization of twins or triplets was according to the sex and birth weight of the firstborn infant. Written informed consent was obtained from mothers. The primary outcome of the DINO trial was neurodevelopment assessed by the Bayley Scales of Infant Development (2nd edition) at 18 months CA. Approval for follow-up of secondary outcomes, specifically growth, body composition, and blood pressure, for infants enrolled in the DINO trial was granted by institutional review boards at each centre.

In the intervention group (high-DHA diet), lactating women were asked to consume six 0.5 g capsules of DHA-rich fish oil per day (900 mg of DHA and 195 mg of EPA to raise the breast milk DHA content to about 1% of total fatty acids. If infant formula was required, a matched preterm formula containing DHA as 1% of total fatty acid was provided. In the control group (standard-DHA diet), lactating women were asked to consume six 0.5 g capsules daily of soy oil (no DHA or EPA). The infants in the control group received DHA at the standard dietary level (~0.3% total fatty acids) from breast milk. If infant formula was required, a matched preterm formula containing 0.3% DHA was provided. All capsules were similar in size, shape, and color and were donated by Clover Corporation, Sydney, Australia. Ready-to-feed preterm infant formula was manufactured to trial specifications by Mead Johnson Nutritionals (Evansville, IN, USA) and packaged to match the capsule containers. Parents, outcome assessors, investigators, and clinicians were blinded to treatment allocation.

### 2.2. DINO 7y Follow-Up Study

All children enrolled in the DINO trial who had not withdrawn consent or died before 7-year CA were eligible for the follow-up study. Written informed consent was obtained from the parent/guardian of the child. Children attended clinic appointments at one of five centres participating in the original DINO trial.

Height and weight, waist, and hip circumference were measured by trained research personnel at each hospital using standardized techniques [14]. Sex and age-specific z-scores for children were calculated according to the WHO Child Growth Standards [19]. Bioelectrical impedance spectroscopy was used to measure lean body and fat mass [20]. The technique provides a measure of total body water used to estimate lean body and fat mass using previously validated equations in pediatric populations [20]. Systolic, diastolic, and mean arterial blood pressure was measured with the child sitting with the right arm in the horizontal position using a DINAMAP Procare V100 monitor (GE Healthcare) with an appropriately sized cuff.

### 2.3. Statistical Analysis

Analyses were performed on an intention-to-treat basis; all children who consented to participate in the 7-year follow-up were analyzed according to the original group to which they were allocated, whether they completed the assessments or not. The primary analysis was based on imputed data and included all participants who consented to the follow-up study. To assess missing outcome data, multiple imputation was performed separately by treatment group using chained equations to create 100 complete datasets under the assumption that data were missing at random [21]. Sensitivity analyses were performed on the available data and imputed data for n = 657 children in the original sample, excluding deaths. All analyses produced similar results, and only the results of the primary analysis are presented.

Normally distributed outcomes were analyzed using linear regression models, with the effect of treatment expressed as a mean difference with 95% confidence intervals (CI). Binary outcomes were analyzed using log-binomial models, with the effect of treatment expressed as a relative risk with 95% CI. All models used generalized estimating equations to account for the clustering of infants within mothers. Both unadjusted and adjusted analyses were performed, with adjustments made for sex and birthweight of the child and study center. Because unadjusted and adjusted analyses did not differ markedly, only adjusted analyses are presented. Secondary analyses were performed to test for evidence of effect modification by the sex and birthweight (<1250 g and ≥1250 g) of the child. Statistical significance was assessed at the two-sided *p* < 0.05. All analyses followed a prespecified statistical analysis plan and were performed using SAS version 9.3 (Cary, NC, USA) and Stata Release 12 (Statacorp LP, College Station, TX, USA).

## 3. Results

### 3.1. Participant Flow and Loss to Follow-Up

Six hundred fifty-seven mothers consented to the DINO trial from 2001 to 2005. From 2008 to 2013, following the exclusion of deaths (n = 18) and withdrawals (n = 13), n = 626 (95%) children and their families were approached to participate in the 7-year follow-up study. Of the 626 children, 22 were lost to follow-up leaving 604 children who participated in the study. Of these, n = 556, n = 569, n = 566, n = 560, n = 498, and n = 526 children had data for weight, height, waist circumference, hip circumference, body fat mass, and blood pressure, respectively (Figure 1).

### 3.2. Baseline and Follow-Up Characteristics

Baseline maternal and neonatal enrolment characteristics of all infants enrolled in the DINO trial have been described elsewhere. For this follow-up study, children were assessed at a mean of 7.2 y CA in both groups. As in the original trial, more mothers had multiple pregnancies in the standard-DHA group than in the high-DHA group (Table 1)**.** The consumption of DHA-containing foods or supplements by children was similar between groups, Table 1.

### 3.3. Anthropometric Outcomes

There was no effect of high-DHA on any anthropometric measures, either raw measures or Z-scores at 7-year CA (Table 2).

There was little evidence of effect modification by child sex for any anthropometric measures at 7-year CA (Table 3).

There were significant interactions between treatment and child birthweight strata (<1250 g and ≥1250 g) for both height and height z-score in children at 7-year CA (Table 4). In subgroup analysis, infants born <1250 g in the high-DHA group were, on average, 1.2 cm shorter than those in the standard-DHA group at 7-year CA. However, this was reversed in infants born >1250 g, with children in the high-DHA group 1.1 cm taller. These post-hoc analyses did not reach statistical significance. 

### 3.4. Weight Status

At 7-year CA, there were no group differences in the percentage of children classified as underweight or overweight. (Table 5). However, children in the high-DHA group were 1.6 times more likely to be classified as obese than those in the standard-DHA group.

### 3.5. Blood Pressure

At 7-year CA, there was no difference in systolic, diastolic, or mean arterial blood pressure between children in the treatment groups (Table 6). There were no significant treatment differences in blood pressure in subgroup analysis by child sex at birth or birthweight strata.

## 4. Discussion

Since the earliest reports of supplementing preterm infants with formulas containing *n*-3 LCPUFA, concerns have been raised about whether growth was compromised [22]. Our DINO 7-year follow-up is the only study to evaluate the effect of supplying the estimated in utero accretion rate of DHA in infancy on growth as assessed by a broad range of anthropomorphic measures. We also examined the data according to randomization strata, including sex and birth weight. We had previously examined these children at birth, and 4, 12, and 18 months CA. There was no effect of our high-DHA intervention on weight, but infants in the high-DHA group were 0·7 (95% CI 0.1,1.4) cm (*p* = 0·02) longer at 18 months CA [7]. These differences had disappeared by 7 years of age. However, we found a significant interaction between birthweight strata and treatment on height at 7 years (P-interaction, 0.03). Children assigned to receive higher DHA with a birthweight < 1250 g were 1.2 cm shorter, while those with a birthweight > 1250 g were 1.1 cm taller. However, in post-hoc analyses the differences did not reach statistical significance and are not consistent with earlier findings based on birth weight strata. It may be that this is a chance effect related to the high numbers of comparisons. The slight difference in the prevalence of obesity (BMI z-score >95th percentile) in the children assigned to the high-DHA group needs to be interpreted cautiously given the small number of children classified as obese and the large number of comparisons. Bioelectrical impedance spectroscopy measurements at 7-year CA showed no effect of higher DHA on lean body mass or fat mass overall or in subgroup analysis. Our findings are consistent with our recent DHA supplementation study of infants born before 29 weeks had no effect on weight and height z-scores at 5 years CA [11]. We could not confirm the findings of the Scottish study where girls who received DHA-supplemented formula were heavier and had greater skin fold thicknesses [16]. However, the Scottish findings should be interpreted cautiously because of bias due to incomplete outcome data with 55% attrition at the 10-year follow-up.

*N*-3 fatty acid supplementation may improve cardiometabolic disease outcomes by lowering blood pressure [23,24]. We could detect no long-term effects of DHA treatment in infancy on blood pressure at 7-year CA. Preterm infants have been suggested to be at greater risk for cardiometabolic disease in later life than those born at term [17]. No differences were found between preterm and term-born adults for most features associated with metabolic syndrome in a recent systematic review, except blood pressure, where adults born preterm had significantly higher systolic and diastolic blood pressure [25]. A systematic review of children born at term showed that the effects of *n*-3 fatty acids on obesity and blood pressure were inconsistent [26]. In the Scottish study [16], DHA supplementation from birth to 9 months CA did not affect blood pressure overall in children at 10 years. However, in contrast to our study, girls responded differently to boys for systolic blood pressure (sex by treatment interaction effect *p* = 0.047). Girls receiving DHA-supplemented formula had significantly higher systolic blood pressure than those receiving the control formula without DHA. The effect was no longer significant once adjusted for current weight.

Our DINO 7-year CA follow-up study is the longest to evaluate the effect of supplying the estimated in utero accretion rate of DHA to preterm infants on anthropometric and cardiometabolic outcomes. The study’s strengths include the sample size, the minimal risk of bias achieved through blinding research personnel, outcome assessors, and participants, and the minimal rate of attrition. The follow-up had clinically relevant prespecified secondary outcomes and a prespecified analysis plan. Our study was a pragmatic trial involving five Australian perinatal centers, suggesting the results’ generalizability to the broader preterm infant population in high-income countries.

## 5. Conclusions

Overall, the DINO 7-year follow-up results show that DHA supplementation early in the life of a preterm infant is unlikely to compromise growth, body composition, or improve blood pressure later in life.

## Figures and Tables

**Figure 1 nutrients-15-00335-f001:**
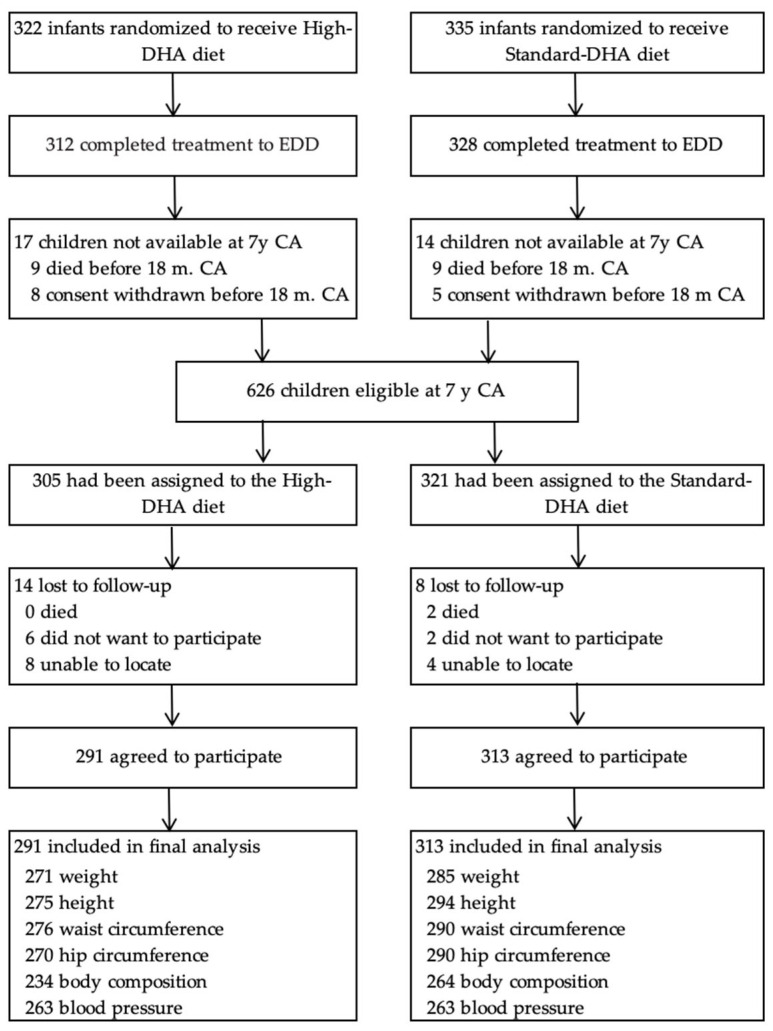
Participant flow and follow-up.

**Table 1 nutrients-15-00335-t001:** Baseline and 7-years follow-up participant characteristics by treatment group ^1^.

	High-DHA (n = 291)	Standard-DHA (n = 313)
Baseline		
Gestational age, weeks	30.0 (28.0, 31.0)	30.0 (28.0, 31.0)
Birth weight, g	1307 ± 420	1320 ± 410
Male, n (%)	152 (52.2)	172 (55.0)
Singleton birth, n (%)	208 (71.5)	195 (62.3)
Birth weight <1250 g	132 (45.4)	139 (44.4)
Birth weight <5th percentile	33 (11.3)	37 (11.8)
Birth length ^2^, cm	38.2 ± 3.9	38.2 ± 4.1
Maternal age, y	29.9 ± 5.7	30.5 ± 5.2
Smoked during pregnancy	77 (26.5)	76 (24.3)
Infant breastmilk at trial entry	271 (93.1)	291 (93.0)
**7-Year Follow-up**		
Corrected age at follow-up ^3^, y	7.2 ± 0.4	7.2 ± 0.3
Fish in the previous month ^4^	248 (87.0)	267 (89.6)
-no. fish meals in the previous month	5.0 ± 3.8	5.1 ± 3.6
DHA-enriched food in the previous month ^4^	172 (60.4)	181 (60.7)
DHA supplements in the previous month ^5^	73 (25.8)	57 (19.5)

^1^ Values are mean ± SD or n (%), except for gestational age, which is median (IQR); ^2^ Data available for n = 223/228 infants; ^3^ Data available for n = 284/293 infants; ^4^ Data available for n = 285/298 infants; ^5^ Data available for n = 283/293 infants.

**Table 2 nutrients-15-00335-t002:** Growth and body composition indicators at 7-year corrected age, by treatment group ^1^.

	High-DHA (n = 291)	Standard-DHA (n = 313)	Adjusted Difference ^2^	*p* Value
Weight, kg	24.3 ± 5.4	23.9 ± 5.0	0.5 (−0.4, 1.4)	0.28
Weight z-score	0.15 ± 1.34	0.06 ± 1.28	0.09 (−0.13, 0.31)	0.41
Height, cm	122.2 ± 6.6	122.2 ± 6.1	0.1 (−1.0, 1.2)	0.86
Height z-score	−0.01 ± 1.14	−0.00 ± 1.10	0.00 (−0.19, 0.19)	1.00
BMI, kg/m^2^	16.1 ± 2.4	15.9 ± 2.4	0.3 (−0.2, 0.7)	0.21
BMI z-scores	0.19 ± 1.26	0.05 ± 1.26	0.15 (−0.07, 0.36)	0.18
WC, cm	56.5 ± 6.6	56.0 ± 6.3	0.6 (−0.5, 1.7)	0.28
WC z-scores	0.40 ± 1.45	0.29 ± 1.39	0.13 (−0.11, 0.37)	0.29
WHC	0.89 ± 0.05	0.89 ± 0.06	0.00 (−0.01, 0.01)	0.60
WHC z scores	0.29 ± 0.87	0.33 ± 1.06	−0.04 (−0.20, 0.13)	0.68
Lean body mass, kg	19.3 ± 2.7	19.3 ± 2.6	0.0 (−0.4, 0.5)	0.86
Fat mass. kg	4.8 ± 3.5	4.3 ± 3.3	0.4 (−0.2, 1.0)	0.17

BMI, Body Mass Index; WC, Waist Circumference; WHC, Waste to hip circumference. ^1^ Data are presented as mean ± SD, with adjusted difference of means (95% CI) as the treatment effect, based on imputed data. ^2^ Adjusted for center, birth weight, and sex; z-scores adjusted for hospital and birth weight only. Reference data for calculating z-scores were based on WHO Growth Standards.

**Table 3 nutrients-15-00335-t003:** Comparison of growth and body composition indicators at 7-year corrected age, stratified by child sex and treatment group ^1^.

	High-DHA (n = 291) ^2^	Standard-DHA (n = 313) ^3^	Adjusted Difference ^4^	P Interaction ^5^	*p* Value
Weight (kg)				0.41	
Female	24.6 ± 5.7	23.7 ± 5.1	0.9 (−0.5, 2.2)		0.20
Male	24.0 ± 5.2	23.9 ± 4.9	0.2 (−1.0, 1.3)		0.79
Weight z-score				0.40	
Female	0.3 ± 1.3	0.1 ± 1.2	0.2 (−0.1, 0.5)		0.24
Male	0.0 ± 1.3	0.0 ± 1.3	0.0 (−0.3, 0.3)		0.98
Height (cm)				0.65	
Female	122.1 ± 6.5	121.8 ± 5.7	0.4 (−1.1, 1.8)		0.64
Male	122.3 ± 6.6	122.5 ± 6.4	−0.1 (−1.6, 1.4)		0.87
Height z-score				0.65	
Female	0.1 ± 1.1	0.0 ± 1.0	0.0 (−0.2, 0.3)		0.75
Male	−0.1 ± 1.1	−0.0 ± 1.2	−0.0 (−0.3, 0.2)		0.76
BMI (kg/m^2^)				0.33	
Female	16.4 ± 2.7	15.9 ± 2.4	0.5 (−0.2, 1.1)		0.14
Male	15.9 ± 2.2	15.9 ± 2.4	0.1 (−0.4, 0.6)		0.75
BMI z-score				0.38	
Female	0.3 ± 1.3	0.1 ± 1.2	0.2 (−0.1, 0.5)		0.12
Male	0.1 ± 1.3	0.0 ± 1.3	0.1 (−0.2, 0.3)		0.69
WC (cm)				0.70	
Female	56.7 ± 7.0	56.0 ± 6.7	0.8 (−0.8, 2.5)		0.33
Male	56.4 ± 6.3	56.0 ± 6.0	0.4 (−1.0, 1.8)		0.56
WC z-score				0.70	
Female	0.5 ± 1.5	0.4 ± 1.4	0.2 (−0.2, 0.5)		0.35
Male	0.3 ± 1.4	0.2 ± 1.4	0.1 (−0.2, 0.4)		0.60
WHC				0.5	
Female	0.9 ± 0.0	0.9 ± 0.1	0.0 (−0.0, 0.0)		0.92
Male	0.9 ± 0.0	0.9 ± 0.1	−0.0 (−0.0, 0.0)		0.43
WHC z-score				0.92	
Female	0.3 ± 0.8	0.3 ± 1.0	0.0 (−0.2, 0.2)		0.86
Male	0.2 ± 0.9	0.3 ± 1.1	−0.1 (−0.3, 0.1)		0.47
Lean body mass, kg				0.74	
Female	18.6 ± 2.5	18.5 ± 2.3	0.1 (−0.4, 0.7)		0.60
Male	20.0 ± 2.7	20.0 ± 2.6	0.0 (−0.6, 0.6)		0.96
Fat mass, kg				0.32	
Female	5.8 ± 3.7	5.0 ± 3.4	0.7 (−0.2, 1.6)		0.12
Male	3.8 ± 3.1	3.7 ± 3.1	0.1 (−0.6, 0.9)		0.69

BMI, Body Mass Index; WC, Waist Circumference; WHC, Waste to hip circumference. ^1^ Data are presented as mean ± SD, with adjusted difference of means (95% CI) as the treatment effect, based on imputed data. Reference data for calculating z-scores were according to the WHO Growth Standards. ^2^ Females n = 139 and Males n = 152; ^3^ Females n = 141 and Males n = 172; ^4^ Adjusted for center and birthweight. ^5^ Interaction child sex by treatment.

**Table 4 nutrients-15-00335-t004:** Comparison of growth and body composition at 7-year corrected age, stratified by child birthweight strata (<1250 g and ≥1250 g) and treatment group ^1^.

	High-DHA (n = 291) ^2^	Standard-DHA (n = 313) ^3^	Adjusted Difference ^4^	P Interaction ^5^	*p* Value
Weight (kg)				0.22	
<1250 g	22.7 ± 5.3	22.9 ± 5.0	−0.1 (−1.4, 1.2)		0.88
≥1250 g	25.6 ± 5.2	24.6 ± 4.8	1.0 (−0.2, 2.1)		0.10
Weight z-score				0.28	
<1250 g	−0.3 ± 1.4	−0.2 ± 1.3	−0.0 (−0.4, 0.3)		0.83
≥1250 g	0.5 ± 1.2	0.3 ± 1.2	0.2 (−0.1, 0.5)		0.15
Height ± cm				0.03	
<1250 g	119.9 ± 6.6	121.1 ± 6.2	−1.2 (−2.8, 0.5)		0.16
≥1250 g	124.1 ± 5.9	123.0 ± 5.8	1.1 (−0.2, 2.5)		0.10
Height z-score				0.04	
<1250 g	−0.4 ± 1.1	−0.2 ± 1.1	−0.2 (−0.5, 0.1)		0.15
≥1250 g	0.3 ± 1.0	0.1 ± 1.0	0.2 (−0.1, 0.4)		0.16
BMI (kg/m^2^)				0.77	
<1250 g	15.7 ± 2.4	15.5 ± 2.4	0.2 (−0.4, 0.8)		0.51
≥1250 g	16.5 ± 2.4	16.2 ± 2.3	0.3 (−0.2, 0.9)		0.25
BMI z-score				0.88	
<1250 g	−0.1 ± 1.3	−0.2 ± 1.2	0.1 (−0.2, 0.4)		0.42
≥1250 g	0.4 ± 1.2	0.3 ± 1.2	0.2 (−0.1, 0.4)		0.26
WC ± cm)				0.63	
<1250 g	55.3 ± 6.5	55.1 ± 6.4	0.3 (−1.3, 1.9)		0.69
≥1250 g	57.6 ± 6.6	56.8 ± 6.2	0.8 (−0.6, 2.2)		0.25
WC z-score				0.65	
<1250 g	0.1 ± 1.4	0.1 ± 1.4	0.1 (−0.3, 0.4)		0.69
≥1250 g	0.6 ± 1.4	0.4 ± 1.4	0.2 (−0.1, 0.5)		0.27
WHC				0.12	
<1250 g	0.9 ± 0.0	0.9 ± 0.1	0.0 (−0.0, 0.0)		0.44
≥1250 g	0.9 ± 0.0	0.9 ± 0.1	−0.0 (−0.0, 0.0)		0.15
WHC z-score				0.14	
<1250 g	0.4 ± 0.9	0.3 ± 1.1	0.1 (−0.1, 0.3)		0.43
≥1250 g	0.2 ± 0.9	0.4 ± 1.0	−0.1 (−0.4, 0.1)		0.20
Lean body mass, kg				0.45	
<1250 g	18.4 ± 2.6	18.6 ± 2.6	−0.1 (−0.8, 0.5)		0.67
≥1250 g	20.1 ± 2.6	19.9 ± 2.4	0.2 (−0.4, 0.8)		0.52
Fat mass, kg				0.22	
<1250 g	4.1 ± 3.4	4.1 ± 3.4	0.0 (−0.8, 0.9)		0.93
≥1250 g	5.3 ± 3.5	4.5 ± 3.2	0.7 (0.0, 1.4)		0.06

BMI, Body Mass Index; WC, Waist Circumference; WHC, Waste to hip circumference. ^1^ Data are presented as mean ± SD, with adjusted difference of means (95% CI) as the treatment effect, based on imputed data. Reference data for calculating z-scores were according to the WHO Growth Standards. ^2^ <1250 g n = 132 and ≥1250 g n = 159. ^3^ <1250 g n = 139 and ≥1250 g n = 174. ^4^ Adjusted for center and sex. ^5^ Interaction child sex by treatment group.

**Table 5 nutrients-15-00335-t005:** Weight status at 7-year corrected age by treatment group ^1^.

	High-DHA (n = 291)	Standard-DHA (n = 313)	Adjusted Relative Risk (95% CI) ^2^	*p* Value
Thin ^3^	31 (10.5)	36 (11.4)	0.9 (0.6, 1.5)	0.74
Overweight (85th to 95th percentile) ^4^	20 (7.0)	25 (8.0)	0.9 (0.5, 1.6)	0.68
Obese (>95th percentile) ^5^	42 (14.4)	28 (9.0)	1.6 (1.0, 2.6)	0.05

^1^ Values are n (%) of children, and treatment effects are relative risks. based on imputed data; ^2^ Relative risks adjusted for center and birthweight; ^3^ Body mass index (BMI) z-score < 10th-ile; ^4^ BMI z-score > 85th-ile; ^5^ BMI z-score > 90th-ile.

**Table 6 nutrients-15-00335-t006:** Comparison of blood pressure measures at 7-year corrected age, by treatment group and by sex at birth and birthweight strata by treatment group ^1^.

	High-DHA (n = 291)	Standard-DHA (n = 313)	Adjusted Difference	P Interaction ^2^	*p* Value
Systolic BP (mm/Hg)	100.3 ± 10.0	101.0 ± 9.8	−0.7 (−2.4, 1.0) ^3^		0.43
Sex				0.53	
Female	100.8 ± 10.8 ^4^	101.1 ± 9.7 ^5^	−0.1 (−2.6, 2.4) ^6^		0.93
Male	99.8 ± 9.2 ^4^	101.0 ± 9.9 ^5^	−1.2 (−3.5, 1.1) ^6^		0.31
Birthweight				0.24	
<1250 g	100.3 ± 9.8 ^7^	100.0 ± 10.5 ^8^	0.4 (−2.1, 2.9) ^9^		0.74
≥1250 g	100.2 ± 10.2 ^7^	101.9 ± 9.2 ^8^	−1.6 (−3.9, 0.7) ^9^		0.17
Diastolic BP (mm/Hg)	57.4 ± 8.4	57.2 ± 7.5	0.4 (−0.9, 1.7) ^3^		0.57
Sex				0.53	
Female	58.3 ± 8.5 ^4^	57.7 ± 7.4 ^5^	0.8 (−1.0, 2.6) ^6^		0.40
Male	56.6 ± 8.3 ^4^	56.7 ± 7.6 ^5^	0.0 (−1.8, 1.9) ^6^		0.97
Birthweight				0.12	
<1250 g	56.0 ± 7.3 ^7^	57.0 ± 7.2 ^8^	−0.7 (−2.5, 1.0) ^9^		0.42
≥1250 g	58.6 ± 9.0 ^7^	57.3 ± 7.7 ^8^	1.3 (−0.6, 3.1) ^9^		0.17
Mean arterial pressure (mm/Hg)	71.7 ± 7.2	71.8 ± 7.1	0.0 (−1.2, 1.2) ^3^		0.97
Sex				0.57	
Female	72.5 ± 7.2 ^4^	72.2 ± 6.9 ^5^	0.5 (−1.1, 2.0) ^6^		0.55
Male	71.0 ± 7.1 ^4^	71.4 ± 7.2 ^5^	−0.4 (−2.1, 1.3) ^6^		0.67
Birthweight				0.57	
<1250 g	70.7 ± 7.0 ^7^	71.3 ± 7.1 ^8^	−0.3 (−2.0, 1.3) ^9^		0.69
≥1250 g	72.5 ± 7.3 ^7^	72.1 ± 7.0 ^8^	0.3 (−1.3, 1.9) ^9^		0.69

BP, Blood Pressure. ^1^ Data are presented as mean ± SD, with adjusted difference of means (95% CI) as the treatment effect, based on imputed data. ^2^ Interaction child sex or birthweight stratum by treatment group. ^3^ Adjusted for center, birthweight, and sex. ^4^ Females n = 139 and Males n = 152. ^5^ Females n = 141 and Males n = 172. ^6^ Adjusted for center and birthweight. ^7^ <1250^.^ g n = 132 and ≥1250 g n = 159. ^8^ <1250 g n = 139 and ≥1250 g n = 174. ^9^ Adjusted for center and sex.

## Data Availability

Deidentified data will be shared with researchers who supply a methodologically sound research proposal following review and approval by the trial steering committee and completion of a signed data access agreement.

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
