# Peer review of "Effect of Docosahexaenoic Acid (DHA) Supplementation of Preterm Infants on Growth, Body Composition, and Blood Pressure at 7-Years Corrected Age: Follow-Up of a Randomized Controlled Trial"

_nutrients, 2023, doi:10.3390/nu15020335_

Round 1
Reviewer 1 Report
The manuscript explores whether and how higher dose DHA supplementation to premature infants would make a difference to several secondary outcomes at 7 years corrected age through 7-year follow-up of preterm infants participating in the DINO trial. The analysis results show that early DHA supplementation of a preterm infant does not affect growth, body composition and blood pressure. And these results are worth popularizing because of their advantages in sample size, risk of bias, rate of attrition and so on. The structure of this manuscript is relatively complete, and the existing statement explains the trail clearly. However, there are still several issues should be solved before publication.
1) The sample size and its changes described in the text seem to be inconsistent with those in Figure 1. For example, there were 18 participants lost due to death according to the text, but only 2 of them could be found apparently in the figure. In a word, the data in Figure 1 needs to be verified or explained more clearly.
2) The statistical data and its analysis results listed in tables are quite clearly, and It will be more intuitive if some statistical results with significant differences are displayed in charts such as bar charts. For example, the significant differences in weight status can be showed in a bar chart.
3) The contents described in the second paragraph of the discussion section are not relevant to the data of this study, and it may be more suitable if these contents were presented in the introduction section to explain the necessity of conducting this study.
4) In the results, the long-term efficacy and safety of DHA supplement should be explained.
5) The citation format of the reference could be unified and corrected.
Author Response
Thank you for your review response attached

Reviewer 2 Report
This text is clear and well structured to reach a large number of readers.
There are only two published studies in preterm infants evaluating outcomes over 2 years, in which high and regular DHA supplementation are compared. One is Scottish and the other is Australian, which is this. It is excellent that this study stands as a comparison, to confirm or not long-term results. Consequently a comparison cannot be an absolute novelty.
It is important, in face of the undoubted benefits of administering DHA on brain development, that possibly effects on growth and blood pressure be ascertained. This is the topic of the study, and meets interest particularly of pediatricians but also of the whole of preterm infants and their parents. This is truly a study that leads to conclusions for so many children, connecting the docs.
In this regard another important point of the manuscript is the clarity of the results as it has been presented, very important from a communicative point of view.
1) There was little evidence of effect modification for anthropometric measures, but in the high-DHA group 1.6 increased risk of obesity 1.6
2) no differences in systolic, diastolic and mean arterial blood pressure were found.
Author Response
thank you. Please see attached
